# Tight Spaces, Tighter Signals: Spatial Constraints as Drivers of Peripheral Myelination

**DOI:** 10.3390/cells14120926

**Published:** 2025-06-18

**Authors:** Luca Bartesaghi, Basilio Giangreco, Vanessa Chiappini, Maria Fernanda Veloz Castillo, Martina Monaco, Jean-Jaques Médard, Giovanna Gambarotta, Marco Agus, Corrado Calì

**Affiliations:** 1Department of Neuroscience “Rita Levi Montalcini”, University of Turin, 10124 Torino, Italy; vanessa.chiappini@unito.it (V.C.); maria.velozcastillo@kaust.edu.sa (M.F.V.C.); 2Neuroscience Institute Cavalieri Ottolenghi, Regione Gonzole 10, 10043 Orbassano, Italy; martina.monaco777@edu.unito.it (M.M.); giovanna.gambarotta@unito.it (G.G.); 3Center for Psychiatric Neuroscience, Department of Psychiatry, Lausanne University Hospital and University of Lausanne (CHUV-UNIL), 1015 Lausanne, Switzerland; basilio.giangreco@chuv.ch; 4Carl Zeiss Microscopy, 73447 Oberkochen, Germany; 5Biological and Environmental Sciences and Engineering Division, King Abdullah University of Science and Technology, Thuwal 23955, Saudi Arabia; 6Cancer Research Center of Lyon (CRCL), Inserm U1052, CNRS UMR 5286, Centre Léon Bérard, Université de Lyon, 69008 Lyon, France; jackmedard@hotmail.com; 7Department of Clinical and Biological Sciences (DSCB), University of Turin, 10043 Orbassano, Italy; 8College of Science and Engineering, Hamad Bin Khalifa University, LAS Building, Doha P.O Box 5825, Qatar; magus@hbku.edu.qa

**Keywords:** peripheral nervous system (PNS), Schwann cells, myelin, neuregulin-1 (NRG-1), in vitro models, microfluidic chambers, “sandwich” technique, mechanotransduction, spatial constraints

## Abstract

Peripheral myelination is driven by the intricate interplay between Schwann cells and axons, coordinated through molecular signaling and the structural organization of their shared environment. While the biochemical regulation of this process has been extensively studied, the influence of spatial architecture and mechanical cues remains poorly understood. Here, we use in vitro co-culture models—featuring microfluidic devices and hydrogel-based scaffolds—to explore how extracellular organization, cellular density, and spatial constraints shape Schwann cell behavior. Our results show that (i) pro-myelinating effects triggered by ascorbic acid administration is distally propagated along axons in Schwann cell-DRG co-cultures, (ii) ascorbic acid modulates Neuregulin-1 expression, (iii) a critical threshold of cellular density is required to support proper Schwann cell differentiation and myelin formation, and (iv) spatial confinement promotes myelination in the absence of ascorbic acid. Together, these findings highlight how spatial and structural parameters regulate the cellular and molecular events underlying peripheral myelination, offering new physiologically relevant models of myelination and opening new avenues for peripheral nerve repair strategies.

## 1. Introduction

Schwann cells (SCs), the principal glia of the peripheral nervous system (PNS), undergo a tightly regulated developmental sequence originating from neural crest progenitors, progressing through SC precursor and immature SC stages before differentiating into either myelinating or non-myelinating (Remak) phenotypes [1]. The maturation process is orchestrated by dynamic interactions among axonal signals, extracellular matrix (ECM) components, and intrinsic transcriptional programs [2]. Throughout development, myelin maintenance, repair, and remyelination, SCs and axons engage in continuous and essential bidirectional communication [3,4]. In early PNS developmental stages, axon-derived signals promote SC survival and proliferation to ensure appropriate matching of SCs and axon numbers. Subsequently, balanced actions of myelination-inhibitory and myelination-stimulating factors control the timing of myelination onset, determining whether SCs commit to becoming myelinating or form Remak bundles [5,6]. In turn, SCs facilitate the precise localization and assembly of multiprotein complexes necessary for myelin internode formation, anchoring myelin to axons and organizing ion channel clusters at and around the nodes of Ranvier [7]. The plasticity of SCs allows them to respond adaptively to injury by entering into a transdifferentiation state that resembles, but is distinct from, their development form. In this injury-induced phenotype, SCs support regeneration by clearing damaged myelin, promoting axonal survival and regrowth, and reactivating the remyelination program once axons reach their targets [6]. Neuron–glia communication is also disrupted in hereditary or acquired PNS disorders like Charcot–Marie–Tooth (CMTs), in diabetic neuropathies, and during aging [8,9,10].

The dialog between glial cells and neurons is mediated through multiple mechanisms: ephaptic communication, exosome-mediated communication, paracrine and juxtacrine signaling, and physical coupling [11]. SCs require apicobasal polarity, established by the presence of the axon on the apical side and the extracellular matrix (ECM) on the basal side [12]. This apicobasal signaling axis ensures the precise spatiotemporal coordination of cytoskeletal reorganization, cell-cycle exit, and myelin sheath formation [4]. Previous studies demonstrated that several membrane-associated proteins expressed at the axon interface, such as Necl-1 and NRG-1, actively contribute to the regulation of myelination [13]. The axonal, membrane-bound isoform of NRG-1 type III plays a central role in myelination by activating different intracellular signaling pathways in SCs. Reduced NRG-1 expression in NRG1^+/−^ mice leads to hypomyelination and decreased nerve conduction velocity, while its overexpression results in hypermyelination [14,15]. Multiple proteases are involved in modulating NRG-1 activity, notably β-secretase 1 (also known as BACE1 or beta-site APP cleaving enzyme 1) and TACE (also known as ADAM17) [4]. These enzymes have opposing effects on NRG-1 function: BACE1-mediated cleavage activates NRG-1 and is essential for proper myelination, as demonstrated by the hypomyelination observed in BACE1-deficient mice [16]. Conversely, cleavage by TACE leads to NRG-1 inactivation; mice with conditional TACE deletion exhibit a hypermyelinated phenotype [17]. On the apical side, SCs express membrane proteins that interact with neuronal counterparts (e.g., NRG-1 receptors ErbB2/ErbB3), while on the basal side, they express receptors that detect ECM components such as GPR-126, dystroglycan, and integrins. Physical signals transmitted through the ECM activate diverse signaling cascades, which also influence actin cytoskeleton dynamics and chromatin organization [18]. Despite its biological significance, the complexity of studying myelination mechanisms in vivo has prompted the development of in vitro models that replicate PNS myelination processes. These models have become invaluable tools for dissecting the intricate signaling pathways, cellular interactions, and molecular dynamics underlying myelin formation and repair. The emergence of 2D and 3D co-culture systems allowed researchers to better mimic the nerve microenvironment, facilitating the study of neurotrophic factor dynamics, myelination, and regenerative mechanisms with increased physiological relevance [19]. More recently, breakthroughs such as human pluripotent stem cell platforms [20], 3D-engineered co-cultures [21], and microfluidic devices [22,23] have revolutionized the field. These advanced models bridge the gap between in vitro and in vivo physiology, allowing for precise investigation of neuro-effector communication, neuroplasticity, and disease mechanisms. They also provide high-throughput platforms for drug screening, enhancing the translational potential of therapies targeting neuropathies and nerve injuries [19].

One of the key factors promoting myelination in vitro is ascorbic acid (AA), which acts through dual mechanisms: epigenetic regulation of pro-myelinating genes and structural stabilization of the ECM. AA facilitates widespread DNA demethylation by serving as a cofactor for Tet enzymes, which convert 5-methylcytosine to 5-hydroxymethylcytosine (5hmC), thereby activating the transcription of myelin-related genes such as Periaxin and Myelin Basic Protein (MBP). Concurrently, AA acts as a cofactor in collagen hydroxylation, supporting basal lamina assembly, a prerequisite for SC polarization and myelin sheath formation [24]. Research on the sodium-dependent vitamin C transporter 2 (SVCT2), the primary transporter of AA in SCs and neurons, has provided critical insights into its role in PNS myelination. In vivo studies using SVCT2 heterozygous knockout mice have revealed significant hypomyelination, ECM abnormalities, and impaired remyelination following nerve injury [25,26]. Despite encouraging results from animal studies, clinical trials investigating AA as a treatment for Charcot–Marie–Tooth disease type 1A (CMT1A) have consistently failed to demonstrate significant therapeutic benefits [27].

Recent research has highlighted the importance of both mechanical and biochemical properties of the ECM as influential factors in myelin formation, with significant implications for tissue engineering applications in the PNS. The cellular responses triggered by mechanical signals transmitted from the ECM to the nucleus are collectively referred to as “mechanotransduction” [12]. Various forms of mechanical cues, such as tension, compression, and changes in ECM composition, differentially influence the expression levels and subcellular localization of transcription factors. The progression of SCs from precursors to mature myelinating cells can be modulated in vitro by physical forces such as compression or substrate stiffness. For instance, the impact of substrate elasticity on SC development has been actively investigated: SCs cultured on soft matrices exhibit an elongated morphology, enriched actin-based processes, low motility, and reduced proliferation, while SCs on rigid matrices adopt a flattened morphology with more stress fibers [28], increasing their motility and proliferation [29]. The expression of SC differentiation markers is unaffected by matrix elasticity alone, but the presence of laminin 211 on stiff substrate activates YAP/TAZ mechanotransducers that translocate into the nucleus [29] and the co-stimulation with cyclic AMP results in Krox20 upregulation, the master regulator of myelin-related gene expression [28]. Substrate stiffness properties also influence dorsal root ganglia (DRG) sensory neurons, modulating neurite outgrowth directionality, growth cone dynamics, and the migration of SC progenitors [30].

Although mechanical and molecular factors are now recognized as crucial for the proper assembly and function of the PNS, the precise nature of the neuronal/axonal and glial signals that initiate and coordinate the spatiotemporal progression of myelination, as well as the specifics of neuron–glia and glia–glia cross-talk, remain largely unresolved. In this study, we established a robust in vitro model to investigate PNS myelination using a dual-compartment microfluidic device. This experimental setup enabled us, for the first time, to describe the intercellular propagation of myelination signals. Further characterization of the process revealed that AA promotes both NRG-1 expression and increased cellular density. Additionally, we demonstrate that the deposition of a hydrogel layer mimicking the basal lamina can trigger myelination in the absence of AA. These findings support the notion that mechanical cues alone are sufficient to initiate myelination in PNS.

## 2. Materials and Methods

### 2.1. Microfluidic Chambers

Microfluidic chambers were assembled following a previously published protocol [31]. Briefly, each chamber consisted of a polydimethylsiloxane (PDMS) structure containing two compartments connected by microchannels, bonded to a glass slide (24 × 50 mm, VWR). The chamber dimensions were as follows: microgroove length = 500 μm, width = 10 μm, height = 3 μm; compartment width = 1.5 mm, height = 1 mm, length = 10 mm; access hole diameter = 6 mm. Relief masters for chamber fabrication were produced via photolithography. The PDMS structure was created using replica molding of Sylgard 184 silicone elastomer (Dow Corning, Midland, MI, USA), polymerized at 70 °C for 1 h. Access holes were generated using a 6 mm biopsy puncher (Stiefel, Research Triangle Park, NC, USA). Both PDMS and glass coverslips were cleaned with water and 70% ethanol before assembly. The two components were then bonded via plasma activation. Finally, the microfluidic chambers were coated overnight at room temperature with 0.1% poly-L-lysine (molecular weight 75,000–150,000; Sigma-Aldrich, Saint Louis, MO, USA) in pure water.

### 2.2. Cell Cultures

DRG were dissected from Sprague Dawley rat embryos at embryonic day 14.5 (E14.5) or from C57/B6 E13.5 mouse embryos as previously described [32]. The dissected DRG were enzymatically dissociated by incubation in 0.25% trypsin (GIBCO, Waltham, MA, USA) at 37 °C for 1 h. Cells were collected via gentle centrifugation (900 rpm for 15 min), resuspended in C-medium (DMEM supplemented with 10% FCS, 1% penicillin-streptomycin (P/S), 200 mM L-glutamine, and 50 μg/mL 2.5S nerve growth factor NGF; Alomone Labs, Jerusalem, Israel), and plated overnight at 37 °C with 5% CO_2_ at a density of 1.5 DRG per 12 mm glass coverslip (VWR International LLC, Radnor, PA, USA) pre-coated with 50 µg/mL ammoniated rat collagen (Cultrex 3440-100-01; R&D Systems, Minneapolis, MN, USA).

For cultures in microfluidic chambers, 2 DRG per chamber were plated in the soma compartment; for DRG neuron–SC co-culture, 1.5 DRG/coverslip were seeded, and for pure DRG neuronal cultures used in protein assays, 5 DRG/coverslip were plated. One hour after plating, the medium was replaced with fresh C-medium, and cultures were incubated overnight at 37 °C. The next day, the medium was replaced with an NB-supplemented medium (Neurobasal medium containing 2 g/L D-glucose, 1% B27 supplement, 1% P/S, 200 mM L-glutamine, and 50 μg/mL 2.5S NGF) and maintained for one week with medium changes every two days. Myelination was induced by supplementing C-medium with 50 μg/mL (284.1 mM) ascorbic acid (Sigma-Aldrich, St. Louis, MO, USA), and cultures were maintained under these conditions for an additional two weeks (for microfluidic chambers) or three weeks (for cultures plated on coverslips), with medium changes every two days.

To obtain pure neuronal cultures, dissociated DRG cultures were plated on pre-coated coverslips with growth factor-reduced Matrigel^®^ (Corning, Corning, NY, USA) and subjected to a 10-day cycling protocol alternating between NB-supplemented medium and NB-supplemented medium containing 10 μM fluorodeoxyuridine (FdU; Sigma-Aldrich, St. Louis, MO, USA). Pure DRG neuronal cultures used in protein assays were seeded on pre-coated coverslips with 0.1% poly-L-lysine (molecular weight 75,000–150,000, Sigma-Aldrich, St. Louis, MO, USA) in pure water.

Primary rat SCs were isolated from sciatic nerves of P3 Sprague Dawley rats, as previously described [33]. Briefly, 20 sciatic nerves were collected, minced, and digested in 0.25% trypsin (Gibco, Waltham, MA, USA) and 0.1% (*w*/*v*) collagenase type IV (Worthington Biochemical Corporation, Lakewood, NJ, USA) for 30 min at 37 °C. Cells were collected by mild centrifugation (2000 rpm for 10 min), resuspended in DMEM supplemented with 10% FCS and 1% P/S, and cultured in 6 cm Petri dishes. Fibroblasts were eliminated by treatment with 10 μM cytosine β-D-arabinofuranoside (ARA-C; Sigma-Aldrich, St. Louis, MO, USA) and subsequent incubation with an anti-Thy 1.1 monoclonal antibody (R&D Systems, Minneapolis, MN, USA) followed by rabbit complement (VWR, Radnor, PA, USA). Purified SCs were expanded in DMEM supplemented with 10% FCS, 1% P/S, 4 mM forskolin (Sigma-Aldrich, St. Louis, MO, USA), and 10 ng/mL human recombinant neuregulin-1 beta 1 (NRG1-β1; R&D Systems, Minneapolis, MN, USA).

To establish DRG neuron–SC co-cultures, primary rat SCs were trypsinized, resuspended in C-medium at a final concentration of 2 × 10^5^ cells per coverslip, and plated onto pure neuronal cultures. Additional chemical compounds used in the experiments included the broad-spectrum ADAM secretase inhibitor GM6001 (364206; Calbiochem, San Diego, CA, USA), the BACE1 inhibitor (565788; Calbiochem, San Diego, CA, USA), Phloretin (524488; Sigma-Aldrich, St. Louis, MO, USA), and Monastrol (M8515; Sigma-Aldrich, St. Louis, MO, USA).

The 3D dome (also known as the “sandwich” technique) was implemented 5 days after plating primary SCs onto neuronal cultures. Then, 200 μL of Growth Factor-Reduced Matrigel^®^ (diluted 1:10; Corning, Corning, NY, USA) was applied over the cultures to create a soft three-dimensional environment, and cultures were maintained for an additional three weeks in the absence of ascorbic acid, with medium changes every two days. An alternative 3D dome method using sodium alginate (SA) was performed as follows: 2% sodium alginate was dissolved in PBS without Ca^2+^ and Mg^2+^ at 60 °C. Once the solution reached 37 °C, the co-culture medium was removed, and 200 μL of the alginate solution was applied directly onto the culture, followed by the addition of 50 μL of 100 mM CaCl_2_ on top of the sodium alginate layer. Gelation was induced by incubation at 37 °C for 5 min. Excess liquid was then aspirated, and cultures were maintained in C-medium without ascorbic acid for three weeks, with medium changes every two days.

### 2.3. Immunohistochemistry

Dissociated DRG cultures or DRG neuron–SC co-cultures were treated as previously described [34]. Briefly, cultures were fixed in 4% paraformaldehyde and permeabilized with cold methanol. Cultures were then blocked in a solution containing 5% normal goat serum (NGS; Vector Laboratories, Newark, CA, USA), 1% bovine serum albumin (BSA; Thermo Fisher Scientific, Waltham, MA, USA), and 0.3% Triton X-100 in PBS for 1 h at room temperature (RT), followed by three washes in PBS (10 min each). Primary antibodies were diluted in blocking solution (2.5% NGS, 0.5% BSA, and 0.15% Triton X-100 in PBS) and incubated overnight at 4 °C. The following day, cultures were washed three times in PBS (10 min each), incubated with the appropriate secondary antibody in PBS, washed again (3 × 10 min), stained with DAPI (1:10,000; Thermo Fisher Scientific, Waltham, MA, USA) for 5 min at RT, washed once more (1 × 10 min), dried, and mounted with Vectashield (Vector Laboratories, Newark, CA, USA). Slides were analyzed using ZEISS Axio Observer Z1 and ZEISS Axio Imager.Z2 microscopes (Carl Zeiss AG, Oberkochen, Germany). The following primary antibodies were used: mouse anti-neurofilament 200 kDa (NF-200; 1:600; N0142; Sigma-Aldrich, St. Louis, MO, USA), rat anti-MBP (1:300; MAB386; Chemicon, San Diego, CA, USA), anti-sodium channel (NaCh; 1:100; S6936; Sigma-Aldrich, St. Louis, MO, USA), anti-Caspr (1:100; 75-001; NeuroMab, Davis, CA, USA), and rabbit anti-phospho-histoneH3 Ser10 (PH3; 1:5000; 06-570; EMD Millipore, Billerica, MA, USA). The following secondary antibodies were used: goat anti-rat IgG (H + L) Alexa Fluor 350, goat anti-mouse IgG (H + L) Alexa Fluor 488, goat anti-rat IgG (H + L) Alexa Fluor 594, goat anti-rabbit IgG (H + L) Alexa Fluor 594, and goat anti-rabbit IgG (H + L) Alexa Fluor 635 (all 1:500; respectively, A21093, A11029, A11007, A11037, and A31576; Invitrogen, Thermo Fisher Scientific, Waltham, MA, USA).

### 2.4. Protein Analysis

Cells were lysed in a buffer containing 95 mM NaCl, 25 mM Tris (pH 7.5), 10 mM EDTA, 2% SDS, 1 mM NaF, 1 mM NaVO_4_, and a protease inhibitor cocktail (1 tablet/10 mL; F. Hoffmann-La Roche AG, Basel, Switzerland). After centrifugation, protein concentration was determined, and 100 μg of total protein lysate was used for standard Western blot procedures. Samples were separated on a 10% SDS-polyacrylamide gel and transferred onto a PVDF membrane (GE Healthcare Technologies, Chicago, IL, USA). Membranes were blocked with 5% non-fat milk in Tris-buffered saline with 0.1% Tween-20 (TBS-T) for 1 h at room temperature (RT) and incubated with primary antibodies overnight at 4 °C and then with secondary antibodies for 45 min at RT. Protein detection was performed using the ECL SuperSignal West Pico (Thermo Fisher Scientific, Waltham, MA, USA) and visualized on Fuji Medical X-ray films or using the Odyssey Infrared Imaging System, with quantification performed using Image Studio Software version 5.5.4 (LI-COR Biosciences, Lincoln, NE, USA). The following primary antibodies were used: chicken anti-MPZ (PZO; 1:10,000; Aves Labs, Davis, CA, USA), mouse anti-α-Tubulin (1:4000; T5168; Sigma-Aldrich, St. Louis, MO, USA), rabbit anti-NRG-1 (1:1000; sc-348; Santa Cruz Biotechnology, Dallas, TX, USA), and mouse anti-β-actin (1:1000; A1978; Sigma-Aldrich, St. Louis, MO, USA). The following secondary antibodies were used: donkey anti-chicken CW800 (1:10,000; 926-32218; LI-COR Biosciences, Lincoln, NE, USA), donkey anti-mouse CW680 (1:10,000; 986-68072; LI-COR Biosciences, Lincoln, NE, USA), anti-rabbit HRP (1:3000; P0399, Dako, Glostrup, Denmark), and anti-mouse HRP (1:3000; P0447, Dako, Glostrup, Denmark).

### 2.5. Replicates, Data Analysis, and Quantifications

All experiments were repeated in triplicate. Data normality was assessed by the Shapiro–Wilk test. All data presented in the graphs showed a normal distribution, except for the datasets in panel 2C, panel 3C (left graph), and panel 3F (left graph). For these latest datasets, since all reads in the untreated samples are always equal to zero, *p*-values were calculated using the one-sample t-test. For all other dataset, *p*-values were calculated using two-tailed unpaired Student’s t-tests. All bar graphs with error bars represent the mean ± SEM. No data points were excluded from the analyses. Differences between two groups were considered statistically significant compared to the controls (untreated) at the following thresholds: *p* < 0.05 (*), *p* < 0.01 (**), *p* < 0.001 (***), and *p* < 0.0001 (****). The statistical details of the experiments are provided in the figure legends. For all experiments, each replicate consisted of samples generated from a pool of four coverslips obtained from each condition in each experiment. For protein analysis, four coverslips per condition per experiment were pooled. In contrast, for immunohistochemistry, each coverslip was analyzed independently.

Quantification of protein analysis was performed using Fiji software version 2.16.0 [35] for chemiluminescence-based membranes and Image Studio Software version 5.5.4 for the LICORbio™ system. For immunohistochemistry experiments, fifteen fields per condition in three independent experiments were analyzed.

The number of nuclei and mitotic cells was determined using the “Detect Cells or Particles” pipeline in the Analysis Panel of Arivis Pro software (version 4.2.1, Arivis AG, Rostock, Germany). This pipeline consists of a sequence of operations designed to analyze images and extract segmented objects. For nuclei analysis, the “Blob Finder—Particles” operation was set to a diameter of 4 μm, a probability threshold of 25%, and a split sensitivity of 45%, based on visual assessment of the segmentation. The same parameters were applied to mitotic cells, except that the diameter was set to 6 μm.

To quantify internodes, Arivis Cloud (Carl Zeiss AG, Oberkochen, Germany) was used to generate an instance segmentation model capable of automatically identifying each internode. Manual segmentation was performed on 20 images representative of the entire dataset to train the model. Several iterative training epochs were conducted to enhance the accuracy of the deep learning algorithm before the final model was uploaded to the “Deep Learning Segmenter” operation in Arivis Pro. The resulting segments were filtered to exclude objects with an area smaller than 13 μm^2^. Each image was uploaded to the software as a separate image set, and the “Batch Analysis” tool was used for processing. To validate the selected methods, nuclei, mitotic cells, and internodes in 18 randomly selected regions of interest (ROIs) were manually counted using Fiji software. No significant differences were observed between the manual counts in Fiji software and the automated results from Arivis Pro, confirming the reliability of the software for automated image analysis.

Image analysis for Neurofilament staining was performed using CellProfiler™ (version 4.2.8), employing a sequential pipeline to enhance, segment, and quantify neurite structures. Neurite detection was improved using the “Enhance-or-Suppress-Features” module with the “Tubeness enhancement” method, which selectively highlights elongated structures. The input images were processed with the following parameters: feature type set to “Neurites,” enhancement method set to “Tubeness,” and a smoothing scale of 1. The enhanced output image was then used for segmentation. Neurite segmentation was conducted using the “Threshold” module, applying the “Adaptive Otsu thresholding” method with three-class classification, where the middle-intensity class was assigned to the foreground. The thresholding method was set to “Otsu” with a correction factor of 1 and a smoothing scale of 1.3488. An adaptive window size of 30 pixels was used, with lower and upper threshold bounds set between 0 and 1.0. The outlier fraction was configured at 0.05 for both lower and upper bounds. This step resulted in a binary image, which was subsequently analyzed for area coverage. The “Measure-Image-Area-Occupied” module was used to quantify the area occupied by the detected neurites. The measurement of the segmented binary image provided the proportion of image pixels occupied by the detected neurites. The extracted quantitative data were exported for further statistical analysis. The proportion of neurite coverage was compared across experimental conditions to assess differences in neurite density.

### 2.6. Animals

Mice and rats were housed under specific-pathogen-free (SPF) conditions in a controlled environment with a 12 h light/dark cycle. They had free access to water and a standard chow diet. Animals were sacrificed by exposure to CO_2_.

## 3. Results

### 3.1. In Vitro Myelination in Dual-Compartment Microfluidic Devices

To gain insight into the molecular pathways involved in the interactions between neurons and associated glial cells, we optimized a neuron–glia co-culture system using compartmentalized microfluidic chambers. These devices are composed of two main compartments for cell growth, connected and communicating with each other through 100 microgrooves: the “soma compartment”, where dissociated DRG are plated, and the “neurites compartment”, where SCs are in contact with projecting axons (Figure 1A). We optimized the microfluidic device geometry and cell seeding conditions to enable the culture of two dissociated DRG in the soma compartment without additional purification steps, thereby establishing DRG sensory neuron–SC microfluidic co-cultures. Only SCs that are in direct contact with axons and thus receive NRG-1 signaling from the axonal surface, survive, proliferate, and migrate along the neurites [36]. Within the first four days in culture, neurons start extending processes through the microgrooves, and SCs (but not neuronal cell bodies due to their large size) migrate along neuronal processes to the adjacent neurite compartment. After 14 days, the neurite compartment becomes fully populated with SCs, and all microgrooves are occupied by neurites and SCs.

AA is commonly used to induce extracellular matrix formation and to promote myelination in vitro [37]. To evaluate its capacity to induce myelination in microfluidic devices, we supplemented C-medium with 50 μg/mL AA and assessed the presence of myelinated fibers 14 days post-treatment. It is important to note that DRG neuron–SC co-cultures maintained in C-medium without AA consistently fail to develop any myelinated fibers. When AA was added to both the soma and neurite compartments, numerous elongated SCs in both compartments exhibited MBP^+^ staining, as confirmed by immunofluorescence (Figure 1B, left panel). Surprisingly, myelinated filaments were detectable on both sides even if AA was added only to one of the two compartments (Figure 1B middle and right panels). We assessed the maturity and proper assembly of myelinated fibers by examining the compartmentalization of the nodes of Ranvier. Immunostaining revealed correct localization of nodal (sodium channel, NaCh) and paranodal (Caspr) markers in sections of myelinated axons within compartments treated with AA, as well as in untreated compartments connected to those receiving AA (Appendix A). Nevertheless, AA diffusion from one compartment to the other can be excluded. Indeed, microfluidic chambers have been demonstrated to be fluidically isolated both in the absence of cells and in the presence of cortical or hippocampal dissociated neurons and oligodendrocytes [31]. To test fluidic isolation under our experimental conditions, we utilized phloretin, a potent inhibitor of AA uptake. It was previously demonstrated that phloretin inhibits myelination by reducing the uptake of AA [26]. As expected, when phloretin and AA were co-administered in the same compartment, myelination was completely inhibited (Figure 1C, right panel). However, when phloretin and AA were added to opposing compartments (one in the soma and the other in the neurite compartment), myelinated fibers were still detectable in the phloretin-containing compartment (Figure 1C, middle and left panels). These findings indicate that fluidic isolation remains stable over the two-week period required for in vitro myelination and suggest that myelination signals, focally triggered by AA administration, can propagate along fibers in SC-DRG co-cultures.

### 3.2. Ascorbic Acid Influences Cellular Density and NRG-1 Expression

Due to the unique architecture of the microfluidic device, we observed that the presence of internodes correlated with areas of higher cellular density (Figure 2A).

To determine if this phenomenon also occurred in cultures plated on coverslips, we established a co-culture system using mouse DRG neurons and rat SCs. This approach allowed for precise control over the initial number of SCs in each experiment (2 × 10^5^ cells per coverslip). To further investigate the mechanism, we utilized inhibitors of two proteases that regulate functional NRG1-type III levels, and consequently myelination levels in vitro, in opposite ways: GM6001, which inhibits ADAM secretases (among them, the tumor necrosis factor-α converting enzyme, TACE), and BACE1 inhibitor IV, which blocks β-secretase (BACE1) activity [38]. As is consistent with previous findings, we confirmed that AA alone or in combination with GM6001 induced robust myelination in co-cultures compared to untreated controls (898.6 ± 68 MBP^+^ segments/mm^2^ for AA, 1635 ± 119.9 MBP^+^ segments/mm^2^ for AA + GM6001, and no MBP^+^ segments detected in untreated co-cultures; both *p*-values < 0.01). However, when BACE1 inhibitor was present, the ability of AA to promote myelination was almost completely impaired with no significant variation from the control (30 ± 19.6 MBP^+^ segments/mm^2^; *p*-value = 0.07, Figure 2B,C). We also tested the effect of both inhibitors, GM6001 and the Bace1 inhibitor, on DRG neuron–SC co-cultures maintained in C-medium without ascorbic acid: under these conditions, no myelinated fibers were ever observed (Appendix A). Western blot analysis confirmed this variation, showing elevated levels of Myelin Protein Zero (MPZ) in co-cultures treated with AA alone or in combination with GM6001, while only a modest increase was observed in the presence of the BACE1 inhibitor (Figure 2D). A similar trend was observed for overall cell density. Induction with AA or AA + GM6001 led to a significant increase in cell numbers compared to untreated controls (AA: 11,511.5 ± 431 DAPI^+^ nuclei/mm^2^; AA + GM6001: 13,035.7 ± 546.2 DAPI^+^ nuclei/mm^2^; untreated control: 7197.4 ± 427.4 DAPI^+^ nuclei/mm^2^; both *p*-values < 0.001). No significant differences were detected in co-cultures treated with BACE1 inhibitor (8500 ± 759.5 DAPI^+^ nuclei/mm^2^; *p*-value = 0.2, Figure 2C). We then investigated whether cell density is a key factor in promoting myelination. To this end, we cultured co-cultures in pro-myelinating medium (containing AA) together with Monastrol, an inhibitor of Eg5, a mitotic kinesin required for spindle formation and mitotic progression. As expected, Monastrol, blocking mitosis, significantly reduced the Schwann cell numbers after 21 days in culture (4296.8 ± 1503.3 DAPI^+^ nuclei/mm^2^), a statistically significant decrease compared to all other conditions, including the control (*p*-value < 0.05). Co-treatment with 50 μM Monastrol and ascorbic acid almost completely abolished myelination (26.2 ± 19.2 MBP^+^ segments/mm^2^), showing a highly significant reduction compared to ascorbic acid alone (*p*-value < 0.001), but not when compared to untreated controls with a *p*-value = 0.08 (Appendix A). We next examined whether AA and its combination with inhibitors influenced neurite density 21 days post-induction. The results correlated with changes observed in cell and internode numbers. Co-cultures treated with AA showed a 1.28-fold increase in the percentage of areas occupied by neurites compared to untreated controls (AA-treated: 25.4% ± 0.5%; control: 19.9% ± 1.7%; *p*-value < 0.05). Co-treatment with GM6001 further enhanced neurite occupancy (1.34-fold change; AA + GM6001: 26.6% ± 1.6%; control: 19.9% ± 1.7%; *p*-value < 0.05), while the combination of AA with BACE1 inhibitor did not significantly alter neurite extension (22.6% ± 2.9%; *p*-value = 0.4 compared to control; Figure 2C).

To further explore the effects of AA, we investigated its influence on purified neuronal cultures. NRG-1, a key regulator of SC development, plays a crucial role in myelination [14,15]. Among its isoforms, type III is primarily involved in SC myelination. We evaluated the impact of different AA concentrations (10, 25, 50 μg/mL) on NRG-1 expression three days after AA treatment in cultured DRG neurons. Western blot analysis revealed a specific expression pattern, with a 135 kDa band corresponding to the full-length NRG-1 pro-protein and other bands at approximately 65 kDa representing its cleaved forms (Figure 2E). The presence of AA increased the expression of the 135 kDa isoform in a dose-dependent manner, with the highest expression observed at 50 μg/mL (Figure 2E).

### 3.3. Spatial Constraint Promotes Myelination In Vitro

In our models, the presence of differentiated SCs into myelinating glia in co-cultures correlates with increased cellular density, both at glial and neuronal levels. We investigated whether mechanical constraints alone could influence glial differentiation and promote myelination in vitro. To address this, we employed the “3D dome” or “sandwich technique,” previously tested in other cellular systems to promote differentiation in vitro [39]. This method involves adding an artificial extracellular matrix layer over a monolayered culture, in this case, DRG neuron–SC co-cultures, using growth factor-reduced Matrigel^®^ or sodium alginate (SA). These materials encapsulate the cultures within two scaffolds. We plated dissociated DRG neurons onto Matrigel^®^ pre-coated coverslips and purified the cultures to obtain neurons. After seeding 2 × 10^5^ SCs per coverslip, we divided the cultures into three conditions: (i) untreated control (negative control), (ii) co-cultures treated with 50 μg/mL ascorbic acid (AA, positive control), and (iii) cultures encapsulated within the Matrigel^®^ 3D dome (Figure 3A). Twenty-one days after Matrigel^®^ 3D dome deposition, without AA supplementation, we observed a significant increase in myelination compared to the unmyelinated control cultures (434.8 ± 122.6 MBP^+^ segments/mm^2^; *p*-value < 0.05) and a robust increase in cell numbers (3D dome: 10,321.6 ± 1145.3 DAPI^+^ nuclei/mm^2^ vs. control: 6390.5 ± 587.6 DAPI^+^ nuclei/mm^2^; *p*-value < 0.05). While myelination was further enhanced by AA treatment (1564.5 ± 231.9 MBP^+^ segments/mm^2^; *p*-value < 0.05), the increase in cell numbers was comparable to that observed in the co-cultures within the Matrigel^®^ 3D dome (11,542.5 ± 1031.3 DAPI^+^ nuclei/mm^2^; *p*-value < 0.01 vs. control) (Figure 3B,C).

Next, we analyzed two additional parameters at the onset of myelination (5 days post-induction) and after 21 days in culture: (1) whether the increase in cell number was associated with a denser neurite network by comparing different culture conditions, and (2) whether cellular proliferation (assessed by quantifying the nuclei positive for the phosphorylated histone H3) occurred simultaneously with neurite density increase. Five days post-induction, proliferation rates were significantly higher in both AA-treated and Matrigel^®^ 3D dome-encapsulated co-cultures (fold changes of 2.3 and 2.5, respectively, compared to control; AA: 63.5 ± 13.3 PH3^+^ nuclei/mm^2^; Matrigel^®^ 3D dome: 69.0 ± 16.1 PH3^+^ nuclei/mm^2^; untreated control: 27.7 ± 5 PH3^+^ nuclei/mm^2^; both *p*-values < 0.05). Although, at this early time point, neurite occupancy areas showed no significant difference between AA-treated (34.9% ± 1.9%) and control cultures (34.1% ± 0.2%) or between Matrigel^®^ 3D dome-encapsulated (32.4% ± 2%) and control cultures (*p*-value = 0.63 and 0.35, respectively). At 21 days post-induction, proliferation rates remained unchanged across all conditions (control: 28.9 ± 7.3 PH3^+^ nuclei/mm^2^, AA-treated: 34.6 ± 14.3 PH3^+^ nuclei/mm^2^, Matrigel^®^ 3D dome: 18 ± 0.9 PH3^+^ nuclei/mm^2^) with no statistically significant differences (AA vs. control: *p*-value = 0.68; Matrigel^®^ 3D dome vs. control: *p*-value = 0.14). Interestingly, while AA significantly increased neurite density at 21 days (control: 31.4% ± 2.2%, AA-treated: 37.2% ± 1.1%; *p*-value < 0.05), no significant difference was detected with the Matrigel^®^ 3D dome compared to the same control (33.8% ± 1.9%, *p*-value = 0.36) (Figure 3D).

To exclude the possibility that any Matrigel^®^ components (e.g., laminins, collagen) triggered myelination, we tested the “sandwich technique” using sodium alginate (SA), a hydrogel widely used in 3D bioprinting and devoid of bioactive macromolecules [40]. Even in this setting, we observed a robust increase in myelination (243.3 ± 34 MBP^+^ segments/mm^2^; *p*-value < 0.05) accompanied by a 1.7-fold increase in cell number compared to the control (SA 3D dome: 9073.1 ± 439.1 DAPI^+^ nuclei/mm^2^ vs. untreated control: 5324.8 ± 1378 DAPI^+^ nuclei/mm^2^; *p*-value < 0.05). These results demonstrate that culture maturation is promoted by the physical properties of the microenvironment rather than by specific bioactive components within the scaffold (Figure 3E,F).

## 4. Discussion and Conclusions

Although the importance of neuron–glia interactions in regulating myelination is well recognized, the molecular mechanisms underlying the initiation of myelination and the specific nature of neuro-glial cross-talk remain largely unknown. To investigate these processes, we optimized a DRG neuron–SC co-culture system using microfluidic chambers and successfully induced myelination in vitro via AA treatment. Leveraging the fluidic isolation of the device, we independently treated each compartment, applying AA selectively either to the compartment containing only SCs and neurites or the compartment also containing neuronal cell bodies. Unexpectedly, we observed robust myelination even in the compartment that did not receive AA, suggesting the presence of transmitted signals capable of influencing glial fate distally and promoting myelination across compartments.

Previous studies using in vitro models, particularly DRG explant cultures and DRG neuron–SC co-cultures, have established that AA is a key mediator of myelination through dual mechanisms: the epigenetic regulation of pro-myelinating genes and the structural stabilization of the ECM [24]. In DRG explant cultures derived from SVCT2^+/−^ mice, impaired AA uptake led to marked hypomyelination. Since this model is a non-cell-specific heterozygous knockout, the contribution of AA specifically on neurons has not been directly investigated [25]. Other studies have shown that AA positively influences immature neuronal morphology during differentiation, promoting neurite outgrowth [41]. To explain the presence of myelinated fibers in the compartment not exposed to AA, we propose two main interpretations: either AA directly stimulates neurons, allowing a signal to propagate axonally to the opposite compartment, or AA acts on SCs, inducing a paracrine signal that travels through the microgrooves to the other side of the device. While we cannot entirely exclude the latter hypothesis, our observation that myelination is closely associated with the axonal microenvironment and local cellular density led us to focus on a novel neuron-dependent role of AA in PNS myelination. Here, we demonstrate that AA modulates NRG-1 expression in purified neurons in a dose-dependent manner and acts synergistically with it to promote myelination in vitro. Although NRG-1 overexpression has been shown to induce hypermyelination in DRG neuron–SC co-cultures when combined with AA [38], to our knowledge, it has not been previously reported that NRG1 alone, in the absence of AA, is sufficient to induce myelination. We speculate that AA may inhibit the formation of soluble NRG1 isoforms known to antagonize myelination [42].

To initiate myelination, SCs progress through several developmental stages, including (i) migration and division along peripheral axons, (ii) intensive proliferation prior to axonal ensheathment to ensure that each SC myelinates a single axonal segment, and (iii) extensive cytoskeletal remodeling and morphological transformation to achieve radial sorting, during which each axon is segregated in a 1:1 ratio by an individual SC [5]. Zebrafish models have shown that the precise timing of SC division is critical for radial sorting and subsequent myelination in the peripheral nervous system. Delays in SC proliferation during development can impair radial sorting and hinder proper axonal ensheathment [43]. In our system, using AA in combination with NRG-1 secretase inhibitors, we observed that myelination levels correlated with overall culture density, including both SC number and neurite outgrowth. This experimental setup allowed us to identify cellular density as a key parameter influencing effective myelination in vitro. To further explore this phenomenon, we tested whether encapsulating the co-cultures within hydrogels mimicking the mechanical properties of the basal lamina could enhance maturation and support myelination.

Discoveries regarding central nervous system (CNS) development are essential to elucidate how spatial constraints, mechanical tension, and substrate stiffness or elasticity influence cellular behavior and determine cell fate and function. For instance, neural stem cells (NSCs) cultured on synthetic hydrogel differentiate into either neurons or glia depending on the substrate stiffness [44]. Similarly, the differentiation of oligodendrocytes from oligodendrocyte progenitor cells (OPCs) is governed by a complex interplay between intrinsic and extrinsic factors. Intrinsic regulators include epigenetic modulators and transcription factors, while extrinsic cues comprise hormones, growth factors, and physical forces such as tension, compression, and substrate rigidity or elasticity [45]. OPCs are highly responsive to their environment: soft substrate (0.4–1 kPa) promotes migration and proliferation, whereas stiffer substrate (1.5–6.5 kPa) enhances differentiation [46]. Additionally, OPC differentiation is density-dependent; spatial constraints can act as mechanical stimuli, promoting differentiation via the actin cytoskeleton and the Linker of Nucleoskeleton and Cytoskeleton (LINC) complex [18,47]. Conversely, two independent studies in mice exposed to hypogravity for 30 days aboard a bio-satellite reported a reduction in myelin sheath thickness across various spinal cord regions [48] and analysis of lumbar ventral/dorsal root–spinal cord transitional zones also revealed a decreased number of MPZ^+^ cells [49], highlighting the impact of mechanical unloading on peripheral myelination.

SCs are highly responsive to their extracellular environment, particularly its mechanical properties. In both physiological and pathological contexts, SCs are exposed to various mechanical stimuli, such as tensile, compressive, and shear stress, which play crucial roles in their behavior during development, myelination, and injury response. These mechanical cues, together with microenvironmental features such as stiffness and topography, are also key design parameters in engineered scaffolds for peripheral nerve repair [50]. In vitro studies have demonstrated that even low levels of mechanical stimulation can trigger SC proliferation independently of axonal cues [51]. Overall, mechanical inputs significantly affect SC adhesion, migration, elongation, and differentiation. SCs are particularly sensitive to substrate stiffness: on stiff substrates, they adopt a bipolar morphology [52] and upregulate pro-myelinating genes such as Oct6 and Krox20 [53]. In parallel, stiff substrates have also been shown to enhance neurite outgrowth in DRG neuron cultures [53,54]. The mechanical properties of the ECM are primarily governed by elastin and collagen fibers, which confer elasticity and tensile strength. In SCs, the main mechanotransducers involved in sensing and transducing mechanical signals from the ECM and the basal side of the cell are the Hippo pathway effectors YAP and TAZ [29,55] and the mechanosensitive ion channels PIEZO1 and PIEZO2 [56]. Laminins, a major component of basal lamina, can substitute for ascorbic acid in promoting in vitro myelination [57], in part by modulating the NRG1–ErbB2/ErbB3 signaling axis to control the timing and extent of myelination [58], and by promoting the transition of SCs to a myelinating state [59]. In this study, we demonstrated that the deposition of 3D domes composed of either Matrigel^®^ or sodium alginate was sufficient to induce myelination in co-cultures, suggesting that spatial constraints and mechanical inputs can drive SC differentiation toward the myelinating state. By comparing early (5 days post-induction) and late (21 days) culture stages, we observed that both AA and physical confinement induce rapid SC proliferation, which preceded noticeable neurite outgrowth. Interestingly, while AA promoted an expansion of neurite coverage, physical restriction via hydrogels did not show a statistically significant effect on neurite extension. We hypothesize that hydrogels exert mechanical pressure on SCs. However, whether this signal is mediated via the actin cytoskeleton and the LINC complex or whether it mimics collagen-like properties, such as elasticity and tensile strength, and facilitates the local confinement of signaling molecules (e.g., laminins), thereby supporting autocrine or paracrine communication, remains to be clarified. Further studies are needed to dissect the underlying mechanisms.

The 3D dome, or sandwich technique, has been applied in other cellular systems: in human pluripotent stem cells (PSCs), the addition of extracellular matrix components has been shown to trigger mechanical signal transduction that promotes epithelial–mesenchymal transition [39]. In tissue engineering, natural polymers (collagen, gelatin, elastin, hyaluronic acid, and alginate) are widely used for regenerative applications both in vitro and in vivo. Among them, alginate gels have shown particular promise in supporting the regeneration of blood vessels, bone, cartilage, muscle, pancreas, liver, and peripheral nerves [60]. Alginate’s role as a chemically inert material makes it ideal for studying passive physical cues in stem cell differentiation. For instance, controlled modulation of alginate gel stiffness demonstrated that soft substrates could induce spontaneous differentiation of murine ESD3 embryonic stem cells into lineage-specific cell types [61]. Moreover, 3D bioprinting approaches using alginate hydrogels have been employed to guide mesenchymal stem cell (MSC) fate: stiffer regions of the construct preferentially supported osteogenesis over adipogenesis [62]. In neural tissue engineering, alginate-based nerve guidance conduits have been shown to promote peripheral nerve regeneration [63,64] as well as functional recovery after spinal cord injury [65]. Furthermore, scaffolds composed of gelatin/alginate hydrogels seeded with rat SCs have been used to model peripheral nerve injury. These 3D bioprinted constructs enhanced cell adhesion, nerve growth factor secretion, and the expression of genes related to nerve regeneration [40].

In recent years, novel strategies for peripheral nerve repair have emerged, incorporating diverse physical stimulation, which have shown promising potential, including scaffold-like artificial nerve guidance grafts and electrical and mechanical stimulation [66]. Spatially confined conduits have been tested to guide regenerating nerves by slowing axonal growth through gradually narrowing space. This approach, activating YAP-mediated mechanotransduction, suppresses neuroinflammation, improves nerve self-organization, and alleviates pain-related behaviors [67]. The application of mechanical stimuli, or mechanotherapy, is achieved normally by acoustic devices. Following peripheral nerve injury, low-intensity ultrasound (LIU) treatment has been shown to enhance recovery by improving multiple aspects of nerve functionality, including increased axonal number and diameter, enhanced myelination, as well as improved nerve conduction velocity (NCV) and compound muscle action potential (CMAP). Notably, several studies have reported that LIU promotes SC proliferation [68]. LIU also exerts a direct effect on neurons, with in vitro application on DRG cultures significantly enhancing neurite outgrowth [69].

To conclude, the findings presented in this study, in particular, the ability of SCs to respond to mechanical forces and the possibility to modulate their state of differentiation dynamically under geometric constraints, are therefore not only relevant for deepening our understanding of PNS biology but may also help to identify mechanistic pathways that could be harnessed for the development of novel therapeutic strategies aimed at improving PNS myelination under pathological conditions.

## Figures and Tables

**Figure 1 cells-14-00926-f001:**
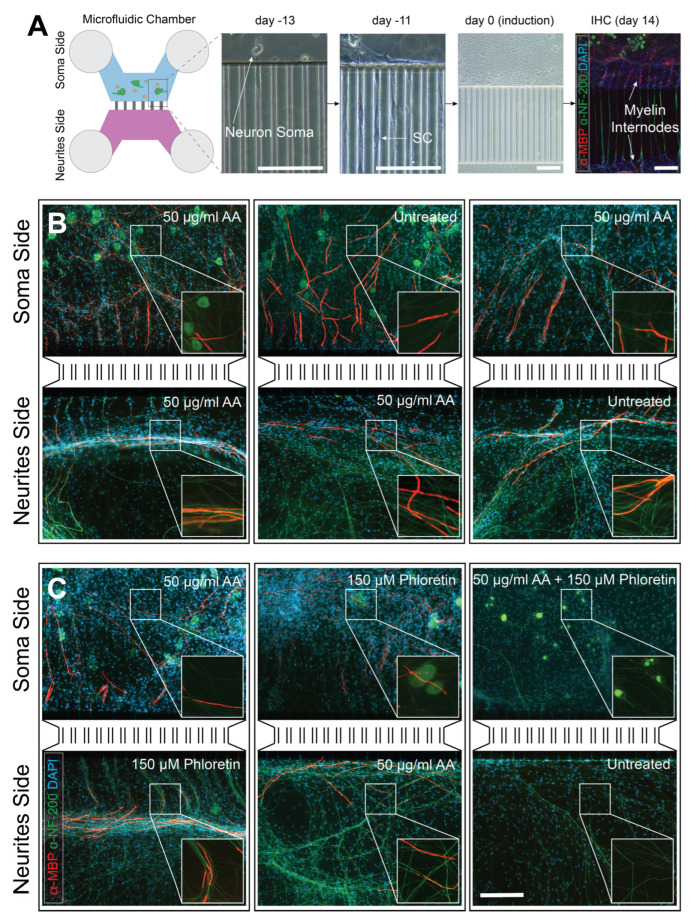
Distal propagation of myelination-trigging signals. (**A**) A schematic representation of a microfluidic dual-compartment device for in vitro modeling of PNS myelination. Dissociated embryonic DRG are seeded in the soma compartment (blue). Between 1 and 3 days after seeding (induction days -13 and -11), DRG neurons extend processes through the microgrooves, and SCs migrate along neuronal processes into the neurite compartment (violet). Within 14 days, both compartments become fully populated by neurites and SCs, with neurites and SCs occupying all microgrooves. Myelination is induced at day 0 by supplementing the culture medium with 50 μg/mL AA for 14 days. Anti-MBP immunostaining confirms the presence of myelin internodes in both compartments. (**B**) Immunohistochemical analysis reveals that myelinated filaments are detectable in both compartments when 50 μg/mL AA is added to one or both compartments, suggesting that the myelination-triggering signal propagates through the microgrooves. (**C**) Immunohistochemical analysis demonstrates that 150 μM phloretin (an inhibitor of AA uptake) impairs myelination only when simultaneously administered with 50 μg/mL AA, but not when the two compounds are added to opposing compartments. This suggests that myelination signals are locally triggered by AA and propagate independently of AA. Green = NF-200 and red = MBP (primary panels and expanded snippets); blue: DAPI (primary panels). Scale bars: (A) Each image has a scale bar = 200 μm, adjusted for the respective magnification. (B-C) Scale bar = 200 μm in panel C applies also for panel B.

**Figure 2 cells-14-00926-f002:**
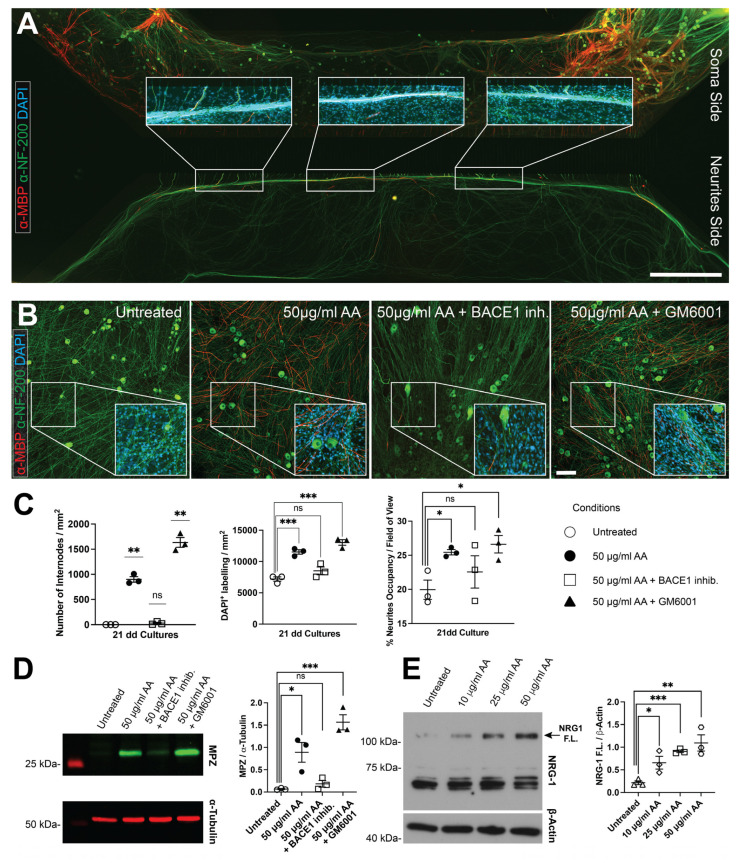
Ascorbic acid enhances cellular density in DRG neuron–SC co-cultures and increases NRG-1 full-length protein expression. (**A**) A multi-frame composite image of a myelinated DRG neuron–SC co-culture grown in a microfluidic dual-compartment device. Magnified insets highlight the presence of myelin filaments in high-cell-density regions. (**B**) Immunohistochemical analysis of DRG neuron–SC co-cultures shows that myelination is triggered in the presence of 50 μg/mL AA or when combined with the ADAM inhibitor GM6001 but is strongly impaired in the presence of a BACE1 inhibitor. (**C**) Quantification of internode numbers, cell counts (nuclei), and neurite network confluency reveals a statistically significant increase in cultures treated with 50 μg/mL AA or in combination with the ADAM inhibitor GM6001, but not in untreated co-cultures or in the presence of a BACE1 inhibitor (statistical analysis refers to the comparison of the different conditions with untreated condition). (**D**) Western blot analysis of co-culture extracts comparing untreated samples and those treated with 50 μg/mL AA, AA in combination with the ADAM inhibitor GM6001, or AA with a BACE1 inhibitor. α-Tubulin was used as a control for equal loading. Relative quantification confirms a significant increase in MPZ expression in cultures treated with 50 μg/mL AA or in combination with GM6001, with only a modest increase observed in the presence of a BACE1 inhibitor. (**E**) Western blot analysis of purified DRG neuron culture extracts treated with increasing concentrations of AA (lane 1 = 0 µg/mL, lane 2 = 10 µg/mL, lane 3 = 25 µg/mL, lane 4 = 50 µg/mL). β-actin was used as a control for equal loading. Relative quantification demonstrates that AA promotes an increase in full-length NRG-1 expression. (**A**,**B**): Green = neurofilament; red = MBP; blue: DAPI. Scale bars: 1 mm (**A**) and 100 μm (**B**). For all quantifications: n = 3; * *p* < 0.05, ** *p* < 0.01, *** *p* < 0.001, ns: not significant; one-sample t-test and Student’s t-test. Data are presented as scatter dot plot with mean ± SEM.

**Figure 3 cells-14-00926-f003:**
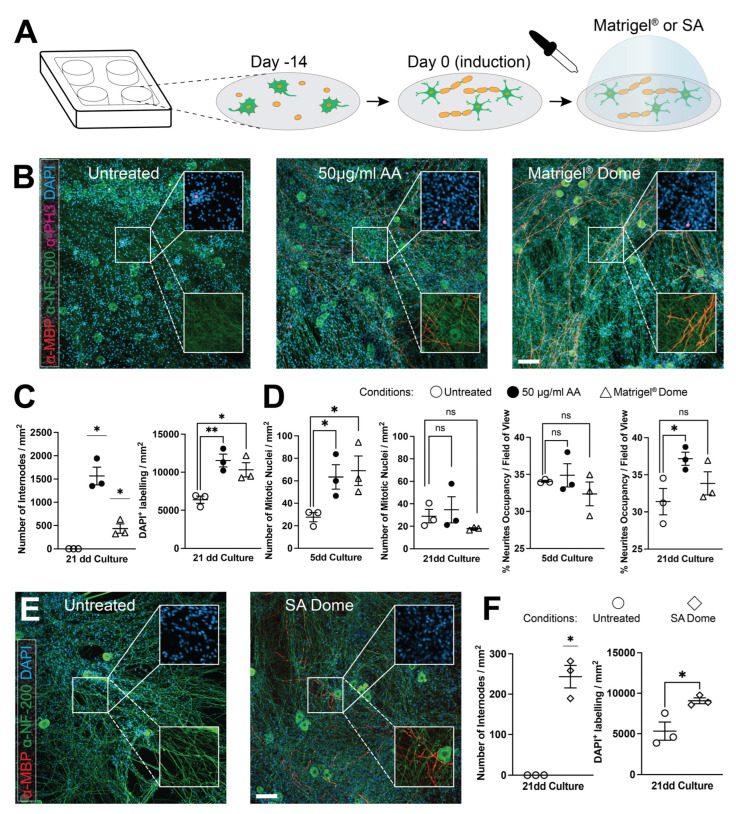
Spatial constraints promote myelination in vitro in the absence of AA. (**A**) A schematic representation of the 3D dome (or sandwich technique) method. DRG-dissociated cultures are seeded onto coverslips and reach confluency after 7 days. The 3D dome is achieved by depositing a layer of gelatinous substrate (Matrigel^®^ or SA) directly on the cell culture monolayer. (**B**) Immunohistochemistry analysis demonstrates that myelination is promoted by encapsulating the co-cultures in Matrigel^®^, even in the absence of AA. (**C**) Quantification of internodes number and cell counts (nuclei) show a statistically significant increase in cultures treated with 50 μg/mL AA or covered with the Matrigel^®^-Dome (statistical analysis refers to the comparison of AA or Matrigel^®^-Dome versus untreated condition). (**D**) Both 50 μg/mL AA stimulation and the Matrigel^®^-Dome promote cell proliferation 5 days after induction, while an increase in neurite network confluency is detectable only in co-cultures treated with AA. (**E**) Immunohistochemical analysis reveals that myelination is also promoted using a 3D dome made with sodium alginate (SA), (**F**) resulting in a statistically significant increase in internode and cell numbers (statistical analysis refers to the comparison of SA-Dome versus untreated condition). (**B**,**E**): Green = NF-200 and red = MBP (primary panels and lower expanded snippets); purple = PH3 and blue = DAPI (primary panels and upper expanded snippets). Scale bars: 100 μm. For all quantifications: n = 3. * *p* < 0.05, ** *p* < 0.01; ns: not significant; one-sample t-test and Student’s t-test. Data are presented as scatter dot plot with mean ± SEM.

## Data Availability

The original contributions presented in this study are included in the article/Appendix A. Further inquiries can be directed to the corresponding author.

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
