# Peer review of "Tight Spaces, Tighter Signals: Spatial Constraints as Drivers of Peripheral Myelination"

_cells, 2025, doi:10.3390/cells14120926_

Round 1
Reviewer 1 Report
Comments and Suggestions for Authors
This appears to be a well conducted study and I have only a couple of minor comments.
Figure 1 - was myelination observed in the absence of both AA and phloretin in both chambers? This would seem to be the one missing important control to say that the transmitted signal following AA supplemented to one chamber was responsible for myelination in the other.
Fig 3C - please confirm / indicate whether the statistical significance is for dome v untreated or dome v AA (or both)
Could the increased myelination under dome conditions be attributed to a transition to a more 3D rather than 2D culture platform rather than changed mechanical forces or spatial constriction?
Some tracked changes remain in the manuscript.
Author Response
We thank the reviewer for assessing our manuscript and her/his constructive criticism. Please find our answer, point by point, as follows:
Comment 1: Figure 1 - was myelination observed in the absence of both AA and phloretin in both chambers? This would seem to be the one missing important control to say that the transmitted signal following AA supplemented to one chamber was responsible for myelination in the other.
Response 1: We thank the reviewer for the comment and we agree that a clarification is needed. We repeatedly tested the condition of culturing without ascorbic acid (AA) and/or phloretin as a negative control, and we can confirm that myelinated internodes were never observed under these conditions. To clarify this point, we have added the following sentence to Section 3.1 at line 344-346:
“It is important to note that DRG neurons-SCs co-cultures maintained in C-medium without AA consistently fail to develop any myelinated fibers (data not shown).”
Comment 2: Fig 3C - please confirm / indicate whether the statistical significance is for dome v untreated or dome v AA (or both)
Response 2: We realized that we didn’t specify well this point. Indeed, statistical significance is for Dome versus untreated and for AA versus untreated. Following reviewer’s question, we specified it better in the Figure 3c legend text at line 540-541:
(c) – […] (statistical analysis refers to the comparison of AA or Matrigel®-Dome versus untreated condition).
And for consistency we also modified the Figure 2(c) legend at line 461-462:
(c) – […] (statistical analysis refers to the comparison of the different conditions with untreated condition).
And Figure 3(f) legend at line 547-548:
(f) – […] (statistical analysis refers to the comparison of SA-Dome versus untreated condition).
Comment 3: Could the increased myelination under dome conditions be attributed to a transition to a more 3D rather than 2D culture platform rather than changed mechanical forces or spatial constriction?
Response 3: While the structure of the setting of the culture is technically 3-dimensional when the 3D-dome is created, the DRG-SCs co-cultures continue to grow in monolayer (2D). We have never observed an extension of the neurites in height. We can then exclude the hypothesis of a change in the spatial properties of the co-culture from 2D to 3D.
Reviewer 2 Report
Comments and Suggestions for Authors
The current manuscript by Bartesaghi et al. brings new insights regarding the myelination process to the discussion. The manuscript is interesting and well-written, the experiments were conducted with great mastery, and the results are strong and support its conclusions. I only have a few amendments to provide:
Minor Amendments:
1. In the panels of Figure 1, I suggest that the authors conduct a colocalization measurement of MBP and NF200 stains to illustrate the interaction between axons and glia better.
2. Could the authors also indicate how many Schwann cells survived in the cultures depicted in Figure 1?
3. In my opinion, providing a graphical abstract or scheme that highlights all your findings would make this work even more attractive to readers. Consider including this suggestion to enhance the scientific soundness of your manuscript.
Author Response
We thank reviewer 2 for her/his assesment of the manuscript, please find our answers below.
Comment 1: In the panels of Figure 1, I suggest that the authors conduct a colocalization measurement of MBP and NF200 stains to illustrate the interaction between axons and glia better.
Response 1: The colocalization of MBP and NF200 observed in our images reflects the expected structural relationship between myelin and axons, consistent with established models of peripheral myelination (“Salzer, J.L. (2015). Schwann cell myelination. Cold Spring Harbor Perspectives in Biology, 7(8), a020529 [DOI:10.1101/cshperspect.a020529]” and “Nave, K.-A., & Werner, H.B. (2014). Myelination of the nervous system: mechanisms and functions. Annual Review of Cell and Developmental Biology, 30, 503–533 [DOI:10.1146/annurev-cellbio-100913-013101]”). Myelin Basic Protein (MBP) is a major component of the myelin sheath, and it is expressed only by Schwann cells ensheathing neuronal axons. We evaluated MBP staining for colocalization with NF-200 and confirmed that all MBP⁺ cells in our model are associated with NF-200⁺ axons. We did not modify the text mentioning this control, but we agree to do it upon request of the reviewer.
Comment 2: Could the authors also indicate how many Schwann cells survived in the cultures depicted in Figure 1?
Response 2: We appreciate the reviewer’s interest in the quantification of Schwann cell survival. While the use of microfluidic chambers provides valuable qualitative insights into PNS biology, we found this setup limited in terms of quantitative analysis. Specifically, although DAPI staining produced clearly visible signals, the image quality was insufficient for reliable cell counting-either manually or with automated tools. For this reason, after initial qualitative observations, we transitioned to traditional co-culture systems on coverslips, which allowed for more robust quantification and analysis.
Comment 3: In my opinion, providing a graphical abstract or scheme that highlights all your findings would make this work even more attractive to readers. Consider including this suggestion to enhance the scientific soundness of your manuscript.
Response 3: We agree with reviewer’s comment and we now included in the manuscript a graphical abstract.
Reviewer 3 Report
Comments and Suggestions for Authors
The study by Bartesaghi and colleagues investigates how spatial organization and mechanical cues influence peripheral myelination. To this end, they use innovative in vitro co-culture systems involving microfluidic devices and hydrogels. While peripheral myelination has traditionally been understood through biochemical signaling between Schwann cells and axons, their findings provide compelling and new evidence that physical and spatial parameters (i.e. cell density and spatial confinement) also play a crucial role in this process.
I believe the work deserves publication. However, I suggest a few revisions/comments that would strengthen the authors' message. Otherwise, I recommend toning down some of the conclusions
Figure 1: to maintain fluidic isolation between the two compartments, a volume difference must be preserved. How was this managed in the ascorbic acid (AA) and phloretin experiments? Is the propagated myelination (MBP amount) quantitatively compared to that in the other compartment?
Is the extent of myelination induced by AA alone comparable to that observed when AA and phloretin are administered to the two separate compartments? Including a quantification of MBP signal could maybe provide more informative insights and strengthen the conclusions (even for other figures).
In figure 2 Has GM6001 treatment alone been performed?
In Figure 3, panel B, it appears that Matrigel Dome alone leads to an increase in MBP synthesis, but not in internode formation. Additionally, Figure 1A states, “We assessed the maturity and proper assembly of myelinated fibers by examining the compartmentalization of the nodes of Ranvier,” and Figure 2C shows a significant increase in the number of internodes following AA treatment. This raises the question: are the myelinating fibers formed after Matrigel Dome deposition not fully mature? If that is the case, I would suggest revising the discussion accordingly, particularly lines 616–619, to reflect this potential limitation.
Lines 572–573: “This experimental setup allowed us to identify cellular density as a key parameter influencing effective myelination in vitro.”
I suggest that the authors tone down this conclusion, as the current data show a correlation between cell density and myelination, but do not establish a causal relationship. To strengthen this point, it would be useful to test whether reducing Schwann cell proliferation, maybe using some antimitotic drugs, impacts myelination and DRG neurite length.
Author Response
We thank the reviewer for the through revision of our manuscript. Find our answers below.
Comment 1: Figure 1: to maintain fluidic isolation between the two compartments, a volume difference must be preserved. How was this managed in the ascorbic acid (AA) and phloretin experiments? Is the propagated myelination (MBP amount) quantitatively compared to that in the other compartment?
Response 1: We thank the reviewer for all valuable comments and suggestions, which have helped us improve the manuscript and clarify our message. The reviewer is correct in pointing out the volume differences between compartments. In the experiments involving only ascorbic acid (AA), the compartment containing AA had always a smaller volume. When two compounds were applied in opposite compartments, we considered it most logical to maintain the smaller volume on the AA side, given its known positive effect on myelination. This strategy helped minimize the risk of false positives. Conversely, we can also exclude the possibility of false negatives, as shown in Figure 1C (left and middle panels), where a strong induction of myelination was observed even when the two compounds were applied in opposite compartments. Regarding MBP quantification, as discussed in the following section, technical limitations make this approach suitable for qualitative but not quantitative analysis. Additionally, there is a significant imbalance in cell density between the two compartments, as only a small fraction of neurites (and the Schwann cells migrating along them) can cross the microgrooves. This discrepancy makes direct comparison between the compartments unfeasible.
Comment 2: Is the extent of myelination induced by AA alone comparable to that observed when AA and phloretin are administered to the two separate compartments? Including a quantification of MBP signal could maybe provide more informative insights and strengthen the conclusions (even for other figures).
Response 2: While microfluidic chambers offer valuable qualitative insights into peripheral nervous system biology, we found this setup limited for quantitative analyses due to several factors, including constraints on high number of replicates and suboptimal staining quality. Therefore, after initial qualitative observations, we transitioned to conventional co-culture systems on coverslips, which enabled more robust quantification and analysis. Unfortunately, this approach did not permit us to perform quantifications as the suggested by the reviewer. Nonetheless, we are confident that, even in the absence of quantitative data, the observations described in Section 3.1 provide sufficient evidence to support our conclusions regarding the distal propagation of the myelination-inducing signal.
Comment 3: In figure 2 Has GM6001 treatment alone been performed?
Response 3: We initially performed this control as part of our investigation into the role of NRG-1 (including also the use of the Bace1 inhibitor in the absence of ascorbic acid), but we did not originally consider including these data in the manuscript. We agree, however, that this is a relevant point, even if the results are negative. Therefore, we have now included an additional figure (see Figure S2A) and added the following sentence to Section 3.2, line 408-411:
“We also tested the effect of both inhibitors, GM6001 and the Bace1 inhibitor, on DRG neurons-SCs co-cultures maintained in C-medium without ascorbic acid: under these conditions, no myelinated fibers were ever observed (Figure S2A).”
Comment 4: In Figure 3, panel B, it appears that Matrigel Dome alone leads to an increase in MBP synthesis, but not in internode formation. Additionally, Figure 1A states, “We assessed the maturity and proper assembly of myelinated fibers by examining the compartmentalization of the nodes of Ranvier,” and Figure 2C shows a significant increase in the number of internodes following AA treatment. This raises the question: are the myelinating fibers formed after Matrigel Dome deposition not fully mature? If that is the case, I would suggest revising the discussion accordingly, particularly lines 616–619, to reflect this potential limitation.
Response 4: Both AA treatment and Matrigel Dome deposition promote myelination, as demonstrated by quantitative analysis of internode number, which shows a significantly higher count in both treated conditions compared to the untreated control (Figure 3). Notably, in the Matrigel Dome condition, staining intensity for both MBP, and also Neurofilament, appears stronger, despite all samples being acquired using identical imaging parameters (e.g., laser intensity, exposure time). We hypothesize that this increased signal intensity may result from the presence of the hydrogel during the immunostaining and imaging processes, potentially influencing the final appearance of the acquired images.
According to our knowledge, the nodes of Ranvier in the PNS are created by a mechanism where specific proteins are clustered at heminodes and then restricted to nodes by paranodal junctions, effectively excluding or compressing proteins from internodal regions after myelin sheath formation (The Nodes of Ranvier: Molecular Assembly and Maintenance. Rasband M.N. and Peles E., 2016. DOI: 10.1101/cshperspect.a020495). The clustering of Nodes of Ranvier is associated with the formation of internodes, supporting the conclusion that the internodes in our cultures likely represent mature myelinated fibers. However, since we did not assess the compartmentalization of the nodes in this experiment, we agreed to revise the aforementioned lines as follows:
“..suggesting that spatial constraints and mechanical inputs can drive SC differentiation toward the myelinating state.”
Comment 5: Lines 572–573: “This experimental setup allowed us to identify cellular density as a key parameter influencing effective myelination in vitro.” I suggest that the authors tone down this conclusion, as the current data show a correlation between cell density and myelination, but do not establish a causal relationship. To strengthen this point, it would be useful to test whether reducing Schwann cell proliferation, maybe using some antimitotic drugs, impacts myelination and DRG neurite length.
Response 5: We once again thank the reviewer for raising this important point. We are pleased to include in the manuscript a set of experiments we had previously performed to explore whether myelination could be mimicked under reduced Schwann cell density, using the mitosis inhibitor Monastrol. These results have been added as Figure S2B and C, and the following sentence has been included in Section 3.2, line 420-429:
“We then investigated whether cell density is a key factor in promoting myelination. To this end, we cultured co-cultures in pro-myelinating medium (containing AA) together with Monastrol, an inhibitor of Eg5, a mitotic kinesin required for spindle formation and mitotic progression.
As expected, Monastrol, blocking mitosis, significantly reduced the Schwann cell numbers after 21 days in culture (4,296.8 ± 1,503.3 DAPI+ nuclei/mm²), a statistically significant decrease compared to all other conditions, including the control (p-value < 0.05). Co-treatment with 50 μM Monastrol and ascorbic acid almost completely abolished myelination (26.2 ± 19.2 MBP+ segments/mm²), showing a highly significant reduction compared to ascorbic acid alone (p-value < 0.001), but not when compared to untreated controls with a p-value = 0.08 (Figure S2B and C).”
Round 2
Reviewer 3 Report
Comments and Suggestions for Authors
I thank the authors for their clear response to my comments. The paper can be accepted